## [Decision Letter · Decision Letter 0]

30 Oct 2025

PONE-D-25-52573Reinforcement Operator Learning (ROL): A Hybrid DeepONet-Guided Reinforcement Learning Framework for Stabilizing the Kuramoto–Sivashinsky EquationPLOS ONE

Dear Dr. Hossain,

Thank you for submitting your manuscript to PLOS ONE. After careful consideration, we feel that it has merit but does not fully meet PLOS ONE’s publication criteria as it currently stands. Therefore, we invite you to submit a revised version of the manuscript that addresses the points raised during the review process.

**ACADEMIC EDITOR:**

This study presents a Reinforcement Operator Learning framework that couples the universal operator approximation of the Deep Operator Network with the online residual learning of the reinforcement learning agent to stabilize the Kuramoto-Sivashinsky equation.

The paper presents interesting and relevant results and could be considered for publication after corrections:

1- The state of the art should be improved, including references from 2025;

2- Present more clearly how the hyperparameter tuning was performed;

3- Include failure or instability cases, exploring where the ROL performs worse;

4- Include a more balanced discussion, highlighting both the strengths and limitations of the proposed framework;

5- Consider including other sections and subsections to describe related works, comparing the results and contribution of this paper with other works.

We look forward to receiving your revised manuscript.

Kind regards,

Angelo Marcelo Tusset

Academic Editor

PLOS ONE

Journal Requirements:

3. We note that your Data Availability Statement is currently as follows: “The authors confirm that the data supporting the findings of this study are available

within the article.”

Additional Editor Comments :

This study presents a Reinforcement Operator Learning framework that couples the universal operator approximation of the Deep Operator Network with the online residual learning of the reinforcement learning agent to stabilize the Kuramoto-Sivashinsky equation.

The paper presents interesting and relevant results and could be considered for publication after corrections:

1- The state of the art should be improved, including references from 2025;

2- Present more clearly how the hyperparameter tuning was performed;

3- Include failure or instability cases, exploring where the ROL performs worse;

4- Include a more balanced discussion, highlighting both the strengths and limitations of the proposed framework;

5- Consider including other sections and subsections to describe related works, comparing the results and contribution of this paper with other works.

Reviewers' comments:

Reviewer's Responses to Questions

**Comments to the Author**

1. Is the manuscript technically sound, and do the data support the conclusions?

Reviewer #1: Yes

Reviewer #2: Partly

Reviewer #3: Yes

2. Has the statistical analysis been performed appropriately and rigorously?

Reviewer #1: Yes

Reviewer #2: Yes

Reviewer #3: Yes

3. Have the authors made all data underlying the findings in their manuscript fully available?

Reviewer #1: Yes

Reviewer #2: Yes

Reviewer #3: Yes

4. Is the manuscript presented in an intelligible fashion and written in standard English?

Reviewer #1: Yes

Reviewer #2: Yes

Reviewer #3: Yes

5. Review Comments to the Author

Reviewer #1: 1- The researcher did not provide any reference to relevant research to clarify the work environment and what has been presented in this field for the past five years until today at least.

2- The research structure needs to be more organized by highlighting the work techniques used and the mechanisms for measuring the efficiency of the proposed system, in addition to the conclusions and their reliance on numerical values for the results.

3- Add at least three references published in 2025.

Reviewer #2: It is highly recommended to consider an other sections for describing the related works so as to help the reader compare your research with recent publications more properly. Moreover, you can consider numbering for sections and subsections.

Reviewer #3: The study proposes an innovative and technically sound hybrid framework. However, kindly please address the following matters;

1. Please explain more how hyperparameter tuning was performed?

2. It would also be valuable to include failure or instability cases, exploring where ROL underperforms (e.g., at higher spatial resolutions or longer horizons). A more balanced discussion highlighting both strengths and limitations would increase the manuscript’s credibility

6. PLOS authors have the option to publish the peer review history of their article (what does this mean?). If published, this will include your full peer review and any attached files.

Reviewer #1: No

Reviewer #2: No

Reviewer #3: **Yes:** Amalina Abdullah

---

## [Author Response · Author response to Decision Letter 1]

19 Dec 2025

Dear Reviewer,

We are very sincerely grateful to you for your careful assessment of our research paper. We have revised our manuscript according to your suggestion and the improvements. We will address the questions raised in your report point by point follows the "Review-Response" file. With the best wishes.

---

## [Decision Letter · Decision Letter 1]

4 Jan 2026

Reinforcement Operator Learning (ROL): A Hybrid DeepONet-Guided Reinforcement Learning Framework for Stabilizing the Kuramoto–Sivashinsky Equation

PONE-D-25-52573R1

Dear Dr. Muhammad Sajjad Hossain,

We’re pleased to inform you that your manuscript has been judged scientifically suitable for publication and will be formally accepted for publication once it meets all outstanding technical requirements.

Kind regards,

Angelo Marcelo Tusset

Academic Editor

PLOS One

Additional Editor Comments (optional):

The authors presented a fully revised version, meeting all the requested corrections and the criteria required for publication of this Journal.

After these considerations, I consider the paper accepted in its current form.

Reviewers' comments:

Reviewer's Responses to Questions

**Comments to the Author**

1. If the authors have adequately addressed your comments raised in a previous round of review and you feel that this manuscript is now acceptable for publication, you may indicate that here to bypass the “Comments to the Author” section, enter your conflict of interest statement in the “Confidential to Editor” section, and submit your "Accept" recommendation.

Reviewer #1: (No Response)

Reviewer #2: All comments have been addressed

2. Is the manuscript technically sound, and do the data support the conclusions?

Reviewer #1: (No Response)

Reviewer #2: Yes

3. Has the statistical analysis been performed appropriately and rigorously?

Reviewer #1: (No Response)

Reviewer #2: Yes

4. Have the authors made all data underlying the findings in their manuscript fully available?

Reviewer #1: (No Response)

Reviewer #2: Yes

5. Is the manuscript presented in an intelligible fashion and written in standard English?

Reviewer #1: (No Response)

Reviewer #2: Yes

6. Review Comments to the Author

Reviewer #1: (No Response)

Reviewer #2: Good Luck.

Thanks a lot for considering the previous comments. However, it is recommended to compare and cite more recent, 2025, publications.

7. PLOS authors have the option to publish the peer review history of their article (what does this mean?). If published, this will include your full peer review and any attached files.

Reviewer #1: No

Reviewer #2: No

---

## [Editor Report · Acceptance letter]

PONE-D-25-52573R1

PLOS One

Dear Dr. Hossain,

I'm pleased to inform you that your manuscript has been deemed suitable for publication in PLOS One. Congratulations! Your manuscript is now being handed over to our production team.

Kind regards,

on behalf of

Professor Angelo Marcelo Tusset

Academic Editor

PLOS One